# Prediction of Non-Uniform Distorted Flows, Effects on Transonic Compressor Using CFD, Regression Analysis and Artificial Neural Networks

**Muhammad Umer Sohail** [1,2,*] **, Hossein Raza Hamdani** [2] **, Asad Islam** [3,4] **, Khalid Parvez** [2] **, Abdul Munem Khan** [2] **, Usman Allauddin** [5] **, Muhammad Khurram** [1] **and Hassan Elahi** [6]

1   Department of Mechanical Engineering, National University of Technology, Islamabad 44000, Pakistan; muhammadkhurram@nutech.edu.pk
2   Department of Aeronautics & Astronautics, Institute of Space Technology, Islamabad 44000, Pakistan; hossein.hamdani@ist.edu.pk (H.R.H.); khalidparvez2009@hotmail.com (K.P.); abdul.munem@ist.edu.pk (A.M.K.)
3   School of Energy and Power Engineering, Beihang University, Beijing 100191, China; asadislam@buaa.edu.cn
4   Department of Mechanical and Aerospace Engineering, Air University, Islamabad 44000, Pakistan
5   Department of Mechanical Engineering, NED University of Engineering & Technology, Karachi 75270, Pakistan; usman.allauddin@neduet.edu.pk
6   Department of Mechanical and Aerospace Engineering, Sapienza University of Rome, 00185 Rome, Italy; hassan.elahi@uniroma1.it
*   Correspondence: umersohail@nutech.edu.pk; Tel.: +92-344-5266-876

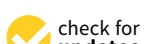



**Featured Application: Development of a transonic compressor instability prediction tool under distorted inlet flow conditions using extensive CFD runs for a supervised learning dataset. ANN with optimal algorithm and different regression learning has been selected to produce all-inclusive transonic compressor rotor performance and behavior at different inlet conditions.**

**Abstract:** Non-uniform inlet flows frequently occur in aircrafts and result in chronological distortions of total temperature and total pressure at the engine inlet. Distorted inlet flow operation of the axial compressor deteriorates aerodynamic performance, which reduces the stall margin and increases blade stress levels, which in turn causes compressor failure. Deep learning is an efficient approach to predict catastrophic compressor failure, and its stability for better performance at minimum computational cost and time. The current research focuses on the development of a transonic compressor instability prediction tool for the comprehensive modeling of axial compressor dynamics. A novel predictive approach founded by an extensive CFD-based dataset for supervised learning has been implemented to predict compressor performance and behavior at different ambient temperatures and flow conditions. Artificial Neural Network-based results accurately predict compressor performance parameters by minimizing the Root Mean Square Error (RMSE) loss function. Computational results show that, as compared to the tip radial pressure distortion, hub radial pressure distortion has improved the stability range of the compressor. Furthermore, the combined effect of pressure distortion with the bulk flow has a qualitative and deteriorator effect on the compressor.

**Keywords:** compressor stall; pressure distortion; swirl flows; stability analysis; CFD; artificial neural networks; regression analysis

## 1. Introduction

For the last few decades, aerodynamic instability in the transonic axial compression system of commercial and military aero engines is under extensive research. Several aircraft engines have witnessed severe operational issues and engine failures due to severe inlet flow distortions. Well-known compressor instability phenomena are rotating stall and surge. Aerodynamic and thermodynamic performances of turbofan engine aircraft

are relying on the flow entrance in compressors. Distortion at the flow entrance creates total pressure non-uniformity at the rotor blades. The non-uniform flow entrance in the compressor may lead to an enormous range of ramifications for the compressor's operability. All commercial and military aircrafts are frequently endangered by complex inlet flow conditions. Surge and rotating stall are undesirable phenomena that cause mechanical, thermal loads, and structural damages to compressor blades, which decrease compressor efficiency and pressure difference. The engine is required to be restarted the in case of an unrecoverable surge, which has catastrophic outcomes in gas turbine engines. These uncertainties may have been generated by operating the compressor ceaselessly away from the surge line. On the other hand, due to its high performance and efficiency, the compressor works close to the surge line. A safety margin should be determined to find the surge avoidance line on the compressor map. However, measures are required when both stall and surge are determined. These conventional control techniques may be active, requiring energy expenditure and control loops, or passive, requiring no auxiliary power and control loops. In both control methods, the compressor characteristic performance map is modified, and the surge line is shifted towards lower mass flow [1].

A turbofan engine is designed to cope up with different climatic conditions, from desert to coastal, tropical, arctic, agricultural, and oil fields. Weather conditions, surrounding temperature, and airborne contaminations have a great influence on the performance of the turbofan engine. The unfavorable effects of non-uniform temperature inlet flow on gas turbine engine operations have always been a hindrance to the performance of turbo-fan engines. Propulsive efficiency is a function of the overall efficiency of the turbofan engine, which itself is dependent on other ambient parameters. The primary concern of distorted inlet temperature is the ingestion of hot gasses from the environment. At high ambient temperatures, air density decreases, reducing the air-fuel mixture for combustion and resulting in a decrease of lift, thrust, and aerodynamic drag. The performance and stability of a transonic axial compressor with non-uniform inlet flow is a significant concern in recent times for its design and operability of a low bypass turbofan engine. In both military and commercial aircrafts, serpentine ducts produce significant inlet swirl distortion. High circumferential swirl flow and inlet flow angularity decrease aerodynamic performance, stall margin, and increase rotor blade loading [2]. Learning algorithms can be helpful in various applications, for example, prediction analysis, clustering, identification of uncertainty, and instability of the data. The objective of deep learning, for the most part, is to comprehend the structure of data and fit that data into models that can be comprehended and used by researchers. The algorithm trains the input data source and utilizes the statistical analysis approach to yield the output values that fall inside a specific range. Based on input data, it develops structure models from sample data to automate the decision-making processes. ML classes depend on how learning is received or how feedback on the learning is given to the framework developed. Sohail et al. [3] investigated a predictive approach based on an Artificial Neural Network (ANN) to predict the transonic compressor performance and behavior at icy, moderate, and extreme hot diversified ambient temperature conditions due to seasonal effects under design RPM. Their model produces substantially accurate results of a compressor rotor at different ambient temperatures when compared with the results of CFD analysis. The results visualized through unity plots are a clear indication that, given any set of temperature and pressure values, the trained model can accurately generate predictions of mass flow rate, temperature ratio, pressure ratio, and efficiency in less computational time, as compared with simulating the models through CFD analysis. However, variable RPM, pressure distortion, bulk flows, and their combined effects on the compressor were not investigated. Furthermore, a shallow Artificial Neural Network model had been developed and trained concerning the same set of features.

Zhong et al. applied reduced-order modeling technology to construct reduced-order models (ROMs) based on the multi-fluid model from CFD data, to simulate biomass rapid pyrolysis in a bubbling fluid bed reactor. CFD calculations were conducted at nine different pyrolysis temperatures. Artificial Neural Network back-propagation was used to map the

species mass fraction data of the CFD simulation onto the pyrolysis temperature and the coordinates of each computational node in the reactor. The number of neurons and the active function of the ANN had been optimized. The ability of the established ROMs to predict species distribution at both training and testing temperatures was investigated [4].

Jiang et al. [5] investigated the centrifugal compressor performance maps of pressure ratio and isentropic efficiency from a shallow dataset using an ANN approach. They developed kriging models based on second-order polynomial and Neural Network models to predict centrifugal compressor performance maps with the limited dataset. Simpers et al. [6] presented an ML regression model to predict the high-pressure hydrogen compressor material and develop a new alloy combination with favorable enthalpies by using an existing open-source database. Their results concluded 28% of mean relative error, whereas due to techno-economic analysis and material cost, composition constrained the new hybrid materials for experimental verification.

Prytz et al. [7] analyzed the new vehicle compressor faults prediction by on-board Volvo truck logged data using ML analysis. Although their results showed that predictive maintenance was possible using the ML approach, due to shallow dataset classification, quality, and cost avoidance, it did not get accurately predicted. The contribution of their ML predictive fault-finding work is highly regarded for identifying the distinctive features of the automobile industry.

In the past, curve fitting, auxiliary coordinates, and non-linear function analytical functions are used for the prediction of the compressor performance map. However, for the mapping approach, scaling and shifting technology have proved more fidelity and accuracy of the gas turbine compressor. Fei et al. [8] proposed Gaussian kernel, feed-forward back-propagation of ANN for prediction of compressor performance map at the ideal conditions with a minimal data sample. Prediction accuracy decreases as the number of samples reduces. Different models were compared in detail with the limited dataset, and it was concluded that the Gaussian kernel function Backpropagation Neural Network (GBPNN) had better prediction performance than the Backpropagation Neural Network (BPNN) and support vector machine (SVM).

Recently, NASA researcher Tong [9] explored the applications of KNN, ANN, and SVM supervised ML approaches to predict the engine core size. For the turbofan engine, machine learning-based predictive tools were developed using the publicly available data of two hundred manufactured engines. The aim was to predict engine core size analysis based on pressure ratio, bypass ratio, and sea-level take-off thrust.

The results showed that the binary classification model predicted core engine sizes with 92% accuracy, whereas the 3-class predictive models, i.e., acceptable, acceptable with improved manufacturing technologies, and unacceptable core sizes, have an overall accuracy of 75%, which predicts undesirable engine core sizes. Ye et al. [10] developed a data-driven method to predict the pressure on a cylindrical body from the velocity distribution in its wake flow. They proposed a deep-learning Neural Network constituted with CNN. The RMSE of the predicted results and the CFD results was less than 0.004. Earlier, a time-marching throughflow method for the off-design performance analysis of axial compressors was analyzed on rotor-67. The method was based on the Euler equations, and an inviscid blade force model was proposed to achieve desired flow deflection. The flow discontinuity problems at the leading and trailing edges were tackled by automatic correction of the blade mean surface using cubic spline interpolation. The flow discontinuity issues at the leading and trailing edges were addressed by using cubic splitting to automatically fix the blade mean surface [11,12].

Earlier authors [13] investigated the flow field in the tip clearance region of the transonic compressor at non-uniform flow conditions under the design and off-design RPM. The results found that hub radial pressure distortion and co-swirl flows improve the stability range of turbofan. Furthermore, compressor performance alleviated the stall at higher RPM combined with distorted inlet pressure.

To the best of the author's knowledge, limited work has been published that covers a hybrid analysis of both deep learning Neural Network and ML regression learners on transonic compressor prediction with multiple inlet variable parameter effects on compressor outcomes. In this work, supervised learning regression learner algorithms and ANN are employed to train data archived from CFD analysis. The current research presents a novel predictive approach based on Regression Learner, ANN, and CFD to predict the effects of pressure ratio, temperature ratio, and mass flow rate distribution on transonic compressor performance. A significant aspect of this work is prediction analysis with high accuracy at different flow conditions with lower computational costs and time. Furthermore, the deep learning Neural Network produces substantially more accurate results when compared with the results of CFD analysis.

## 2. Materials and Methods

Initially, a computational fluid dynamics study was conducted with an ISA model for engine performance comparison with extreme ambient temperature conditions and inlet flow distortion variables. Further calculations were done for dehydrated air, where there is no considerable change in humidity. The adopted pressure changes were kept the same as for the standard atmosphere [14].

Later on, based on an extensive CFD database, a supervised learning feed-forward network topology ANN model has been applied at different compressor inlet conditions, such as pressure distortion, temperature distortion, RPM variations, inlet swirl flow, and a combination of all inlet distortion parameters.

The characteristic map of an axial flow compressor shows the difference in total pressure ratio around the compressor as a function of corrected mass flow at a sequence of constant corrected speed lines. Flow surge or choking may occur if the compressor is not operating within its compressor map range. Inlet enthalpy and pressure ratio both influence the blade speed ratio. The decrease in mass flow compared with the working line causes a higher-pressure rise, and as a result, a larger increase in density in the first stage than was expected at design time. Due to the greater rise in density, the second stage's flow coefficient is much smaller than the first stage, implying an even greater increase in density [15–17].

The literature shows [18,19] that component maps are commonly used in gas turbine output models to describe engine component efficiency in the operating regime. In this research dataset based on CFD analysis, the average on the surface is derived into 6-independent variables i.e., static pressure, RPM, ambient temperature, total pressure, and flow angularity at axial and radial flow, that are required to predict compressor performance in terms of four dependent variables: mass flow rate, pressure ratio, temperature ratio, and isentropic efficiency. Different regression models are used to examine the relationship between dependent and independent variables. Regression statistics facilitates prediction capability and can be utilized to foresee dependent variables when the independent variables are known. The ML-based regression model repository is used to demonstrate the best suited and optimized regression model against the simulation results.

### 2.1. Computational Fluid Dynamics Setup

#### 2.1.1. Mesh Method

Three curve files have been used to create the geometry of the hub, blade, and shroud according to the specification of the geometry, as given in Table 1. Automatic Topology and Meshing (ATM Optimized) is selected for high-quality mesh, and there is no need for a control point of adjustment. ANSYS-Turbo-Grid® and CFX software was used to generate high-quality hexahedral meshes and obtain the CFD solution for the rotor's steady-state case, keeping in view that five different meshes with different grid sizes were generated to study solution dependence upon mesh refinement. 3D mesh at coarse, medium, fine, superfine, and very-fine states were generated. For grid analysis, computations were carried out with 0.37 million to 1.47 million mesh nodes. The grid points were gradated

towards the blade surface, the leading edge, the trailing edge, the hub, the rotor blade tip, and the housing to ensure that the y+ computed stayed below 100 (between 30–100). This value of y+ was within the usual range for the implementation of standard wall functions. It indicated a fair grid resolution near the walls, as the flow solution was assumed to be periodic across the blade row. Table 2 shows the details of the meshes used in the mesh dependence study, and Figure 1 shows the hexahedral-grid mesh with 0.97 million mesh elements, which is a reasonable compromise between computational and experimental results.

**Table 1.** Specification of axial compressor rotor blade [20].

| | |
|---|---|
| Rotor inlet hub-tip diameter ratio | 0.75 |
| Rotor blade Inlet tip radius | 257 mm |
| Rotor blade aspect ratio | 1.56 |
| Tip clearance | 1.016 mm |
| Tip solidity | 1.29 |
| Hub solidity | 3.11 |

**Table 2.** Mesh dependence study [14].

| Sr. No. | Mesh | No. of Nodes (Million) | Design Mass Flow Rate (Kg/S) |
|---|---|---|---|
| 1 | Coarse | 0.037 | 33.43 |
| 2 | Medium | 0.16 | 33.64 |
| 3 | Fine | 0.43 | 33.54 |
| 4 | Super Fine | 0.97 | 33.14 |
| 5 | Very Fine | 1.47 | 33.14 |

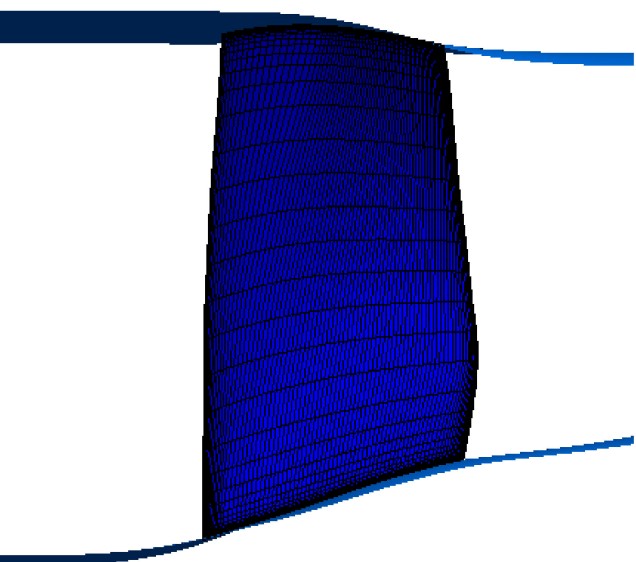

**Figure 1.** Hexahedral grid mesh of a single blade for rotor-67 [13].

### 2.1.2. Clean Inlet Flow Boundary Conditions

Three-dimensional steady compressible Reynolds-averaged Navier-Stoke equations have been solved using the k-$\varepsilon$ turbulence model. At the rotor inlet and outlet boundary, the P-total inlet P-static outlet is considered. Furthermore, the flow direction is specified for a clean inlet at design RPM 16,043, whereas no-slip conditions are used on walls, and periodic conditions are applied at the periodic surfaces. Numerical computations were carried out from choking conditions to near stall conditions by gradually rising outlet

average static pressure to obtain a compressor rotor characteristic map. The near stall point is predicted at the last stable condition of rotor 67. The following equation of Sutherland viscosity law with three coefficients for ideal gas has been applied [21].

$$\mu = \mu o \left( \frac{T}{To} \right)^{\frac{3}{2}} \left( \frac{To + S}{T + S} \right) \tag{1}$$

### 2.1.3. Distorted Inlet Flow Boundary Conditions

Total pressure has radial distribution in hub and tip, in such a way that the total pressure at the blade tip area becomes equal to the total pressure of clean flow inlet in hub radial pressure distortion. Whereas, in tip radial pressure distortion, total pressure at the blade hub area becomes equal to the clean flow inlet at the blade hub area and linearly decreases toward the blade tip area. For hub radial and tip radial inlet pressure, the distortion boundary condition is specified by an expression, as given in Equation (2) [13].

$$P_o \times \left[ \frac{y' + (n - y') \times (r_s - r)}{(r_s - r_H)} \right] \tag{2}$$

Here, $y'$, $r_s$ and $r_H$ are distorted radii at the desired span of the blade. The Inlet boundary condition is changed accordingly to the Equation (3) for co-swirl flow, counter-swirl flow, and flow angularity, pressure distortion, combined pressure, temperature, and bulk flow distortion at desired circumferential or cartesian coordinates [13].

$$\left\{ \begin{array}{c} V_z = cos\alpha \\ V_r = 0 \\ V_{\theta} = V_z \times \tan(\pm\alpha) \end{array} \right\} \tag{3}$$

### 2.2. Deep Learning and Neural Network

Adaptive learning, real-time operation, self-organization, and fault tolerance via redundant information coding are the significant advantages of Neural Networks. For selecting deep learning algorithms, the building blocks are categorized as network topologies, adjustment of learning, selection of an algorithm, and activation function. In this section, an Artificial Neural Network model has been developed and trained for the same set of features required to perform the compressor simulation using CFD analysis, as shown in Figure 2. The model accurately predicts values of mass flow rate, pressure ratio, temperature ratio, and isentropic efficiency at the output by minimizing the Mean Square Error (MSE) loss function. In the test phase, the trained model is evaluated with the test data set. The results obtained from the test data are compared with the simulation results of the CFD analysis. Based on the comparison, unity plots showing the difference between the output and predicted values have been drawn, as stated in Section 3.2.1.

### 2.2.1. Artificial Neural Networks Methodology

This section discusses the methodology used to design, develop, and train a Neural Network model that can generate predictions based on the simulation data of compressor Rotor 67. The way available data is categorized into training, testing, and validation subsets may have a big impact on an Artificial Neural Network's results (ANN). Despite numerous studies, no systematic approach to the best data division for ANN models has been developed, whereas the literature shows that the optimal division of data for Neural Network models is mostly taken as 80% [22–25]. The input and the target value of the Neural Network are set, as shown above in Figure 2.

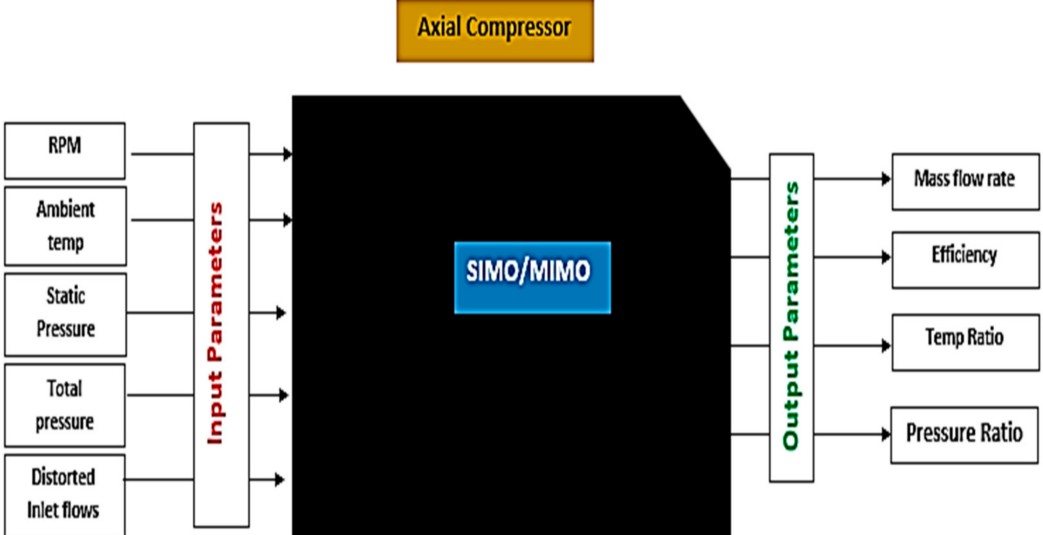

**Figure 2.** Black box for compressor input and output variables.

Dataset Division: There are a total of 2541 samples at different inlet conditions for the Neural Network model. To learn and better model the underlying distribution of input data, the samples have been split so that 80% of the simulation data comprise training data. Table 3 shows the dataset division and the number of samples that fall into each category.

**Table 3.** Dataset division.

| Sr. No. | Dataset | Dataset Division | Total Samples |
|---------|---------|------------------|---------------|
| 1 | Training Data | 80% | 2033 |
| 2 | Testing Data | 10% | 254 |
| 3 | Validating Data | 10% | 254 |

Evaluation Metric: the aim is to perform prediction based on the input set of features; thus, prediction analysis provokes the network to use MSE as an evaluation metric. It is defined as

$$MSE = \frac{1}{n} \sum_{j=1}^{n} (y_j - \hat{y}_j)^2 \tag{4}$$

where $n$ is the total number of samples; $y_j$ is the target value corresponding to a particular set of temperature and ambient pressure values; and $\hat{y}_j$ is the output value, which is obtained through training of the Neural Networks. Model Architecture and Hyperparameters: the designed Neural Network model is based on an Artificial Neural Network's core building blocks, as described by Ian Goodfellow [26]. Table 4 shows the model architecture and hyperparameters.

**Table 4.** Neural Network architecture and hyperparameters.

| Network Design | Model |
|----------------|-------|
| Hidden Layers | 3 |
| Learning Rate | 0.01 |
| Batch Size | 32 |
| Hidden Layers Non-linearity | ReLU |
| Drop out | 0 |

Training of Neural Network: Training refers to estimating a network's parameters (weights and biases) such that it minimizes a cost function. The cost function used is a regression loss, as defined in Equation (4). The network learns the parameters from training

data comprising of input and target feature vectors. The parameters are trained by iterative minimization of the cost function. During the training phase, the ADAM optimizer has been used to update the network's weights and biases. The weights are initialized from a random normal distribution. The initial learning rate value is set to 0.01, and $\beta_1$ and $\beta_2$ values are set to 0.9 and 0.999, respectively [27]. For a generalization purpose, the training data is randomly shuffled and standardized to zero mean and unit variance, while the network has been trained for 2000 epochs.

### 2.2.2. Artificial Neural Networks Setup

Many famous and open-source deep learning frameworks exist, such as TensorFlow, Theano, Torch, and Keras [28–30]. After a brief pre-study of different deep-learning libraries, Keras has been chosen as a framework for implementing neural networks. A preference towards Python programming language eliminated some of the possible deep learning frameworks, such as Torch. As mentioned in Table 4, the model architecture comprises the input layer, two hidden layers, and an output layer; thus, the chosen library is a popular framework for rapid prototyping and developing high-level modularity of Neural Networks.

## 3. Results

As stated above the methodologies, this section is divided into three subsections that are based on CFD results, Artificial Neural Network analysis, and regression-based analysis. Later on, a comparative analysis of ANN and regression learners has been conducted.

### 3.1. Computational Results and Analysis

Figure 3 shows blade flow path geometry, blade mesh, characteristic map validation by plotting pressure ratio versus normalized mass flow rate and the compressor's isentropic efficiency versus normalized mass flow rate under design RPM and tip clearance (TC) of the rotor. Due to axisymmetric surfaces, compressor rotor geometry is defined by distribution points in the meridional Z-R plane, where "Z" represents axial location, and "R" shows the radius of the point on the given surface.

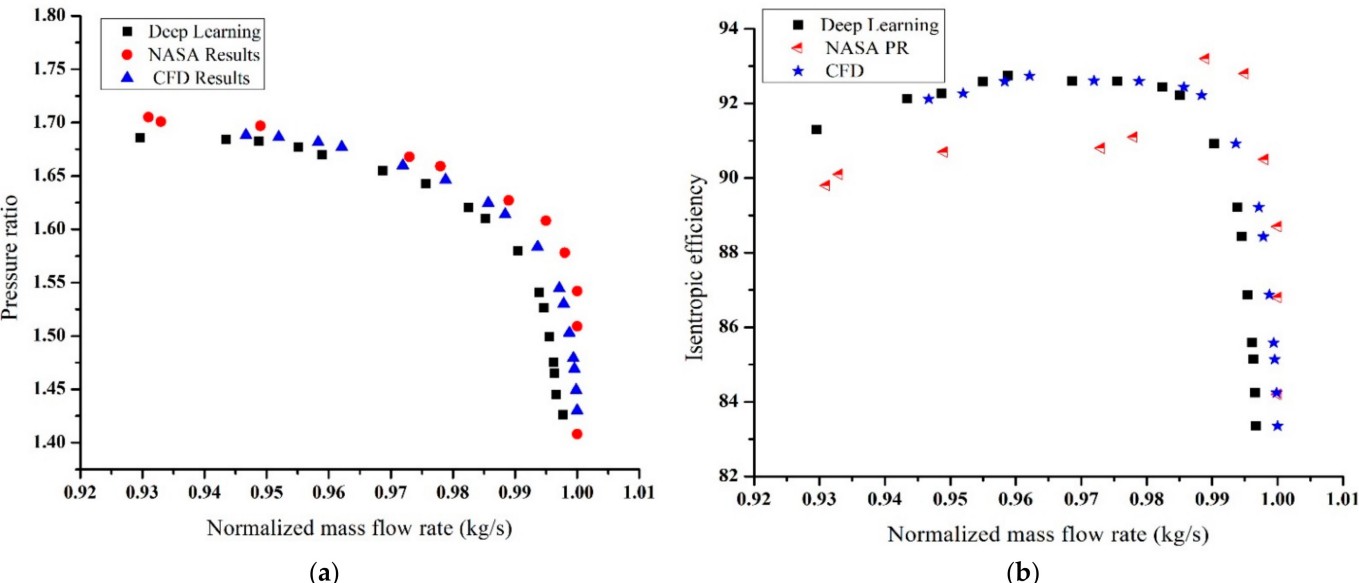

(**a**)　　　　　　　　　　　　　(**b**)

**Figure 3.** (**a**) Pressure ratio vs. normalized mass flow rate, and (**b**) isentropic efficiency vs. normalized mass flow rate.

Figure 4a,b compares the total pressure and temperature ratio loading distribution at peak efficiency. It is observed that the computational total pressure loading is slightly decreased near the leading-edge region. The downstream spanwise variations of total pressure and total temperature ratios depict nice validation with available experimental data.

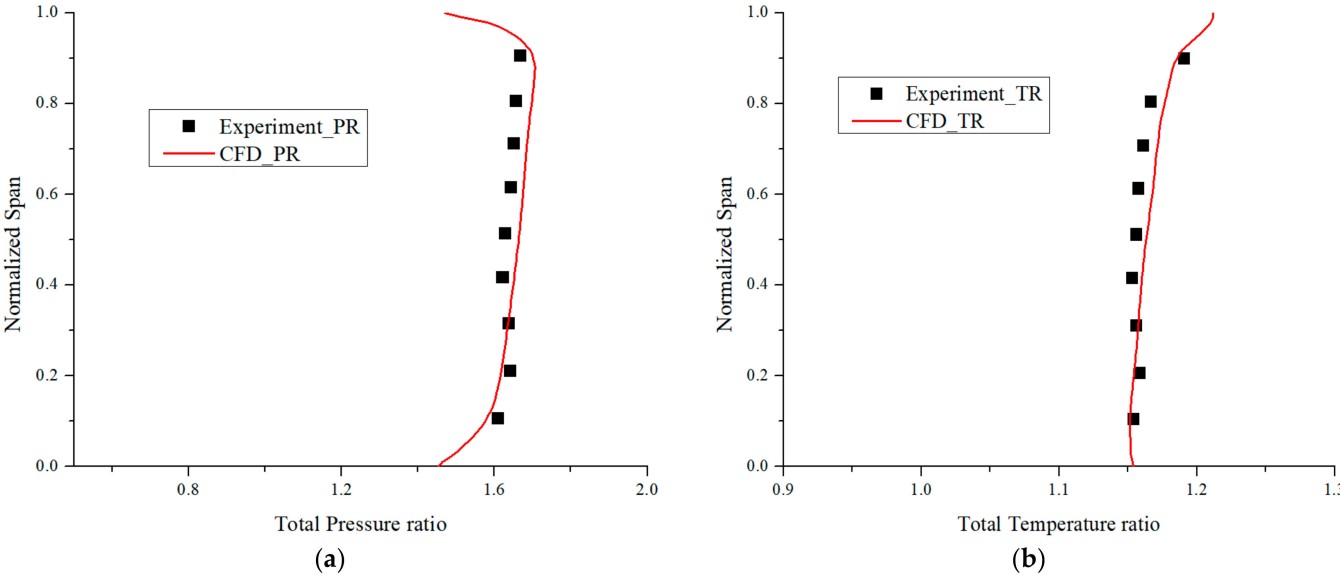

**Figure 4.** Computational and experimental span-wise profiles of (**a**) total pressure ratio and (**b**) total temperature ratio at the peak efficiency condition.

Distorted Inlet Flow Conditions

Non-uniform axial inlet flow frequently occurs in aircraft gas turbine engines that result in chronological distortions of aircraft compressors. High circumferential swirl flow and inlet flow angularity decrease the aerodynamic performance [2]. Distorted inlet flows always have a negative effect on compressor performance and stability, mostly when they are operated at a high ambient temperature and reduced RPM as they deteriorate the compressor rotor. Figure 5a–e shows the characteristic map of hub radial pressure distortion with variable ambient temperature, tip radial pressure distortion with variable ambient temperature, bulk flow with the variable RPM, the combined effects of hub radial pressure distortion with bulk swirl, and tip radial pressure distortion with the bulk swirl on compressor rotor performance and behavior.

The results show that the rotor mass flow rate, efficiency, pressure ratio, and temperature ratio were significantly reduced when the ambient temperature increased excessively. Shallow ambient temperature and denser air produced a higher mass flow rate, pressure ratio, and greater compressor efficiency. The mass flow rate at 99% hub radial and tip radial pressure distortion at 270K had high choking mass flow rates due to low-temperature regions. Still, its stability range was less than the other distorted flows. By further analyzing the characteristic maps of Figure 5a,b, the rotor's stability range is high at 270K of 95% hub and tip radial pressure distortion.

In contrast, it is further analyzed that the hub radial pressure distortion has a better overall stability range than the tip radial distortion at lower ambient temperatures. Figure 5c shows that when the compressor is run at a speed slower than the design speed for various operational reasons, the pressure ratio decreases. In contrast, the operation is at a lower mass flow. Hence the blades are operating at high positive incidence, which may result in a stall. The characteristic maps show that counter swirl flow distortion with higher RPM has higher mass flow rates than the co-swirl flows, but its stability range is lesser than the co-swirl flows.

Furthermore, 10° co-swirl at 90% of design RPM has a better overall stability range than other flow distortion. Hub radial and co-swirl flows have a better comprehensive stability range than tip radial and counter swirl flows. When these distortions are combined, it had a qualitative effect on compressor performance. Figure 5d–g show hub and tip radial pressure distortion combined with bulk flows. The characteristic map shows that at 90%, hub radial combined with 10° co-swirl has a better overall stability range.

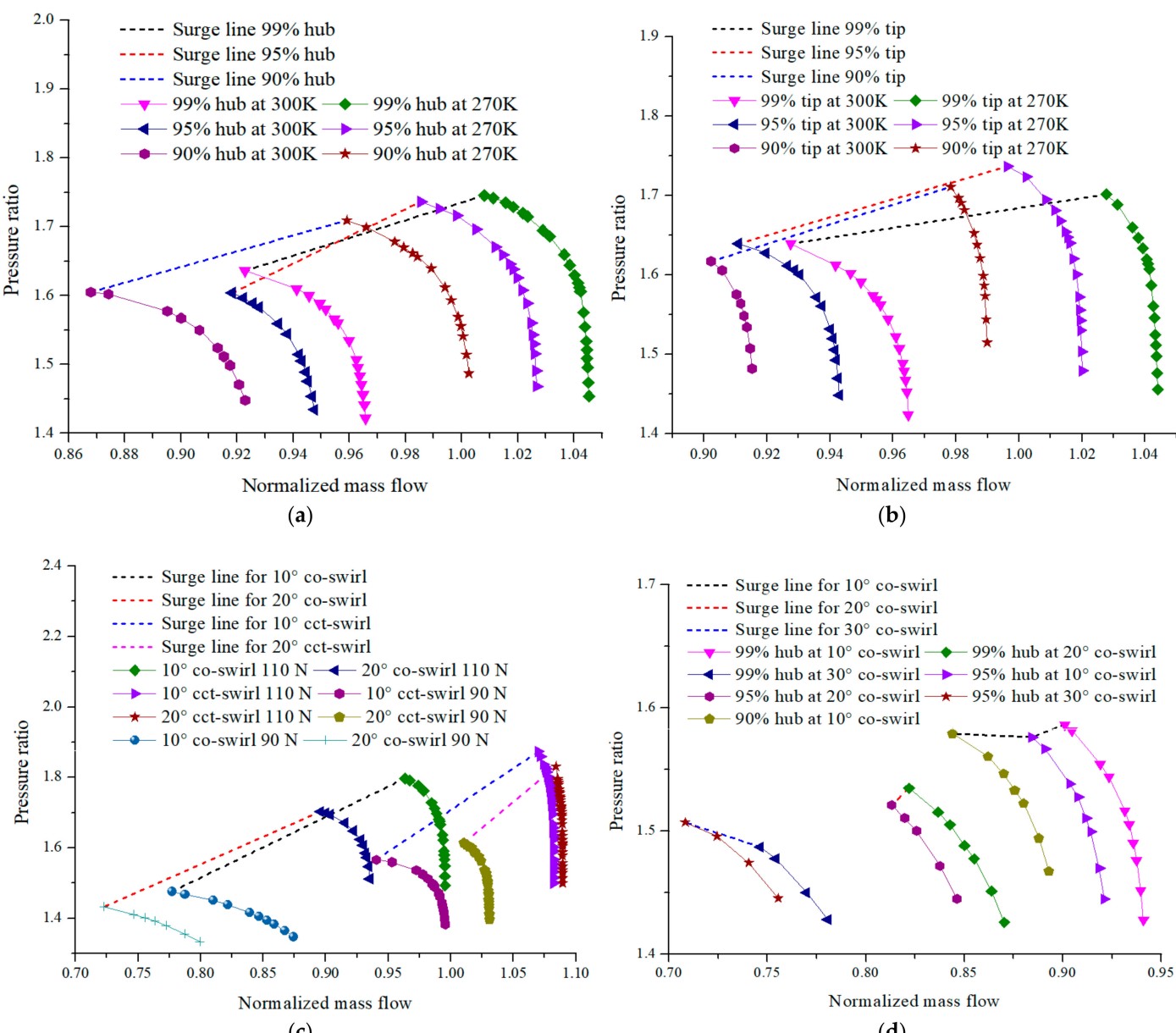

**Figure 5.** *Cont.*

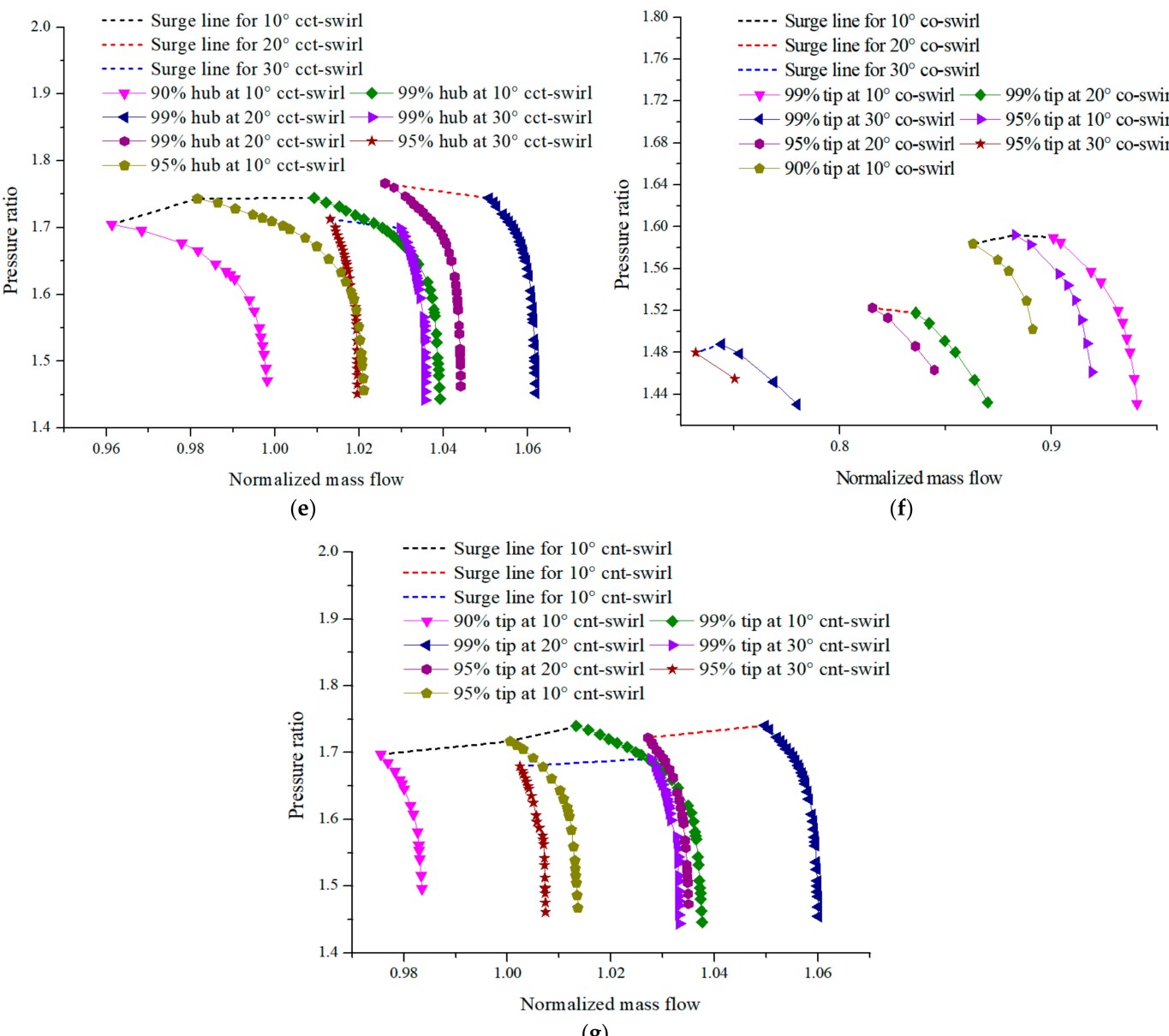

**Figure 5.** Characteristic map of (**a**) hub radial pressure distortion with variable ambient temperature, (**b**) tip radial pressure distortion with variable ambient temperature, (**c**) bulk flow with the variable RPM, (**d**) the combined effects of hub radial pressure distortion with the co-swirl flow, (**e**) the combined effects of hub radial pressure distortion with the counter swirl flow, (**f**) tip radial pressure distortion with the co-swirl flow, and (**g**) tip radial pressure distortion with the counter swirl flow.

### 3.2. ANN Results and Discussions

Figure 6 shows the MSE loss function of the model during the train and test phase. The result indicates that the loss function settles down during the phase of training. The trained model is further tested with the test data, which shows that the loss function follows a complete training data loss function, whereas the loss function remained the same, and therefore the model does not need further training of data.

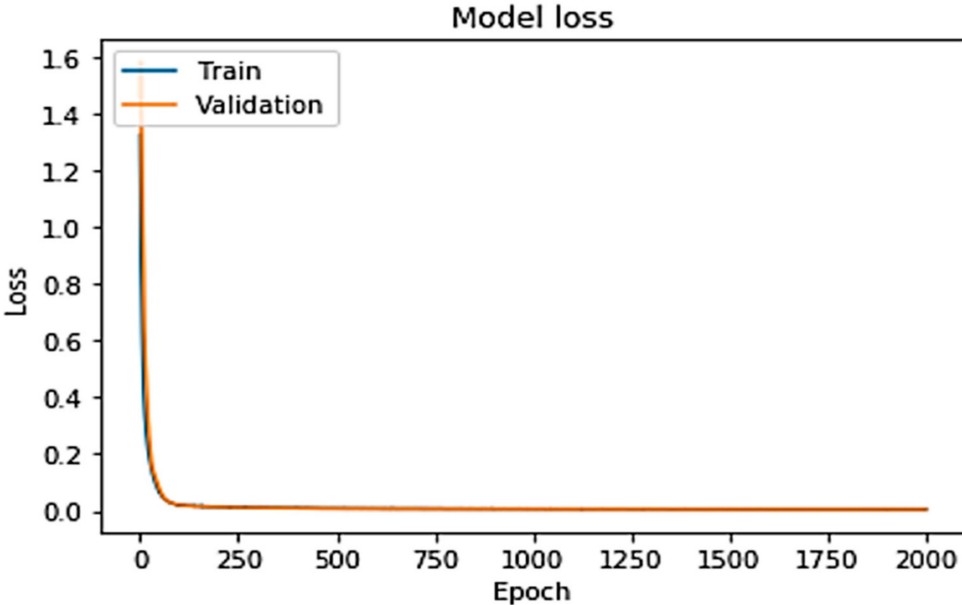

**Figure 6.** Loss function of the Neural Network model during train and test phase.

Table 5 shows the MSE value during train and test time. The values signify generalization of the trained model on the test dataset, as the weight is quite close to the train data MSE value.

**Table 5.** Mean Square Error value at validation and test phase.

| Dataset Division. | MSE Value |
|---|---|
| Validation | 0.024 |
| Testing | 0.029 |

3.2.1. Unity Plots

Earlier a time-marching throughflow method for the off-design performance analysis of axial compressors was analyzed on rotor-67. The method was based on the Euler equations, and an inviscid blade force model was proposed to achieve desired flow deflection. The flow discontinuity problems at the leading and trailing edges were tackled by automatic correction of blade mean surface using cubic spline interpolation. The flow discontinuity issues at the leading and trailing edges were addressed by using cubic splitting to automatically fix the blade mean surface [11,12]. In this research, multilayers, Feedforward, and a supervised learning Neural Network with the ReLU activation function are selected for training, testing, and validating the data set. The unity plot elaborates on the network's performance when the trained model is passed through the test dataset. The plot shows the trained model's generalizability such that the predicted samples should lie closer to the unity line. Figure 7a–d shows the test data unity plots of mass flow rate, pressure ratio, temperature ratio, and isentropic efficiency. The x-axis corresponds to the test dataset's target values, while the y-axis shows the predicted values obtained when the trained model is tested with the test data set.

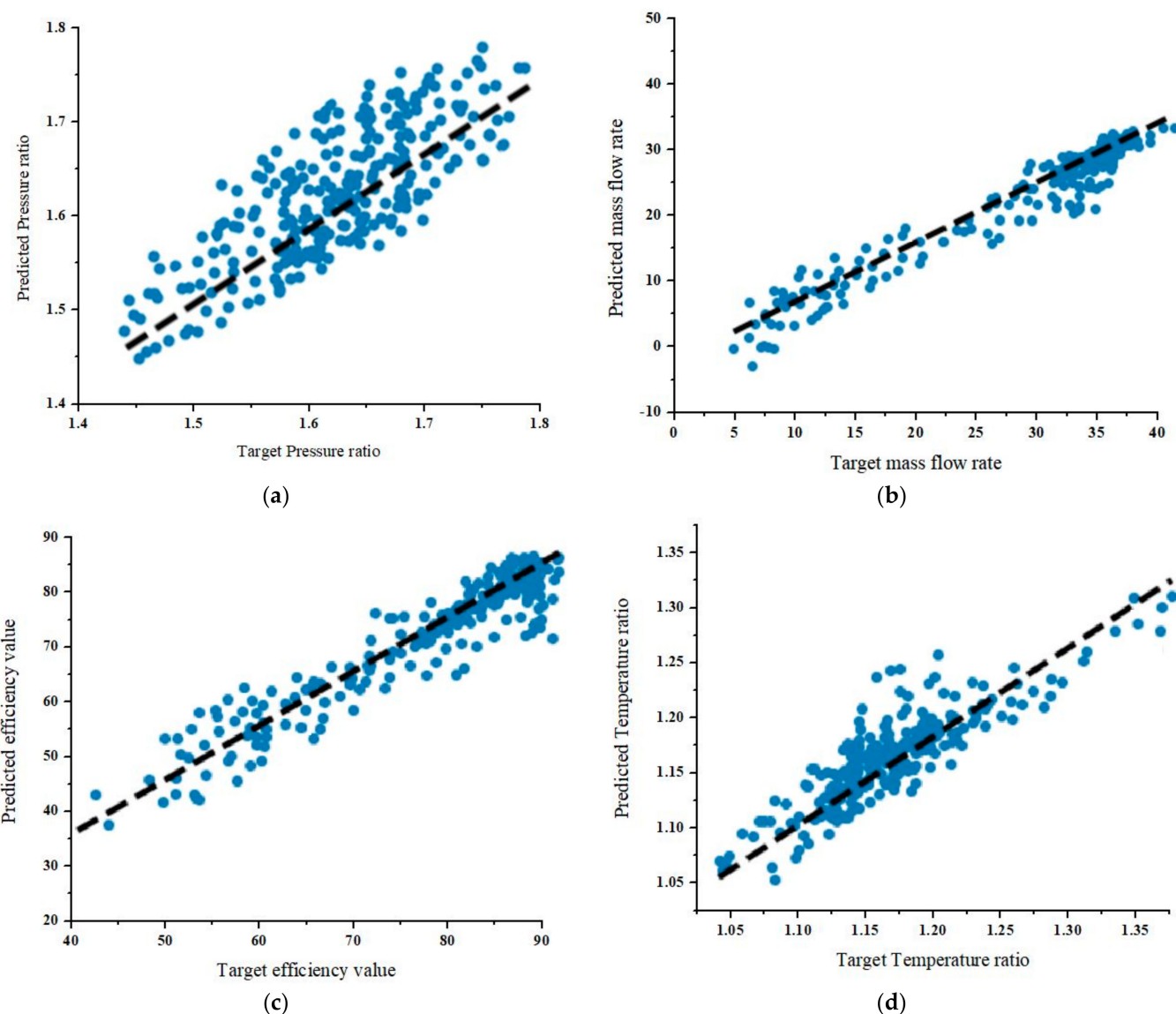

**Figure 7.** Testing dataset unity plots for (**a**) mass flow rate, (**b**) pressure ratio, (**c**) temperature ratio, and (**d**) isentropic efficiency.

### 3.3. Regression Learner Predicted vs. Actual Response

The predicted versus actual regression learner plots are utilized to check model execution after preparing a model. An ideal regression model has an expected response equivalent to the true response, so all the values lie on a regression line. Variations from regression on the line are termed as prediction errors for that point. A reasonable model has minimum errors; thus, the forecasts dissipate close to the regression line. Typically, almost perfect modeled results are evenly spread around the regression line. The complete algorithm of linear regression, stepwise regression, tree (fine, medium, and coarse), SVM (quadratic, cubic, fine, medium, and coarse), ensembled (boosted tree, and bagged tree), and gaussian process regression (GPR) models (squared exponential GPR, Matern 5/2 GPR, exponential GPR, and rational quadratic GPR) are analyzed, whereas the response plot of these models is selected based on predicted versus actual results, which dissipate close to the regression line [31,32]. A comprehensive examination of these models was evaluated in terms of root mean square error and R-squared error.

Based on the minimum error, Figures 8–12 show various regression model learners and their selection for mass flow rate, pressure ratio, temperature ratio, and isentropic efficiency.

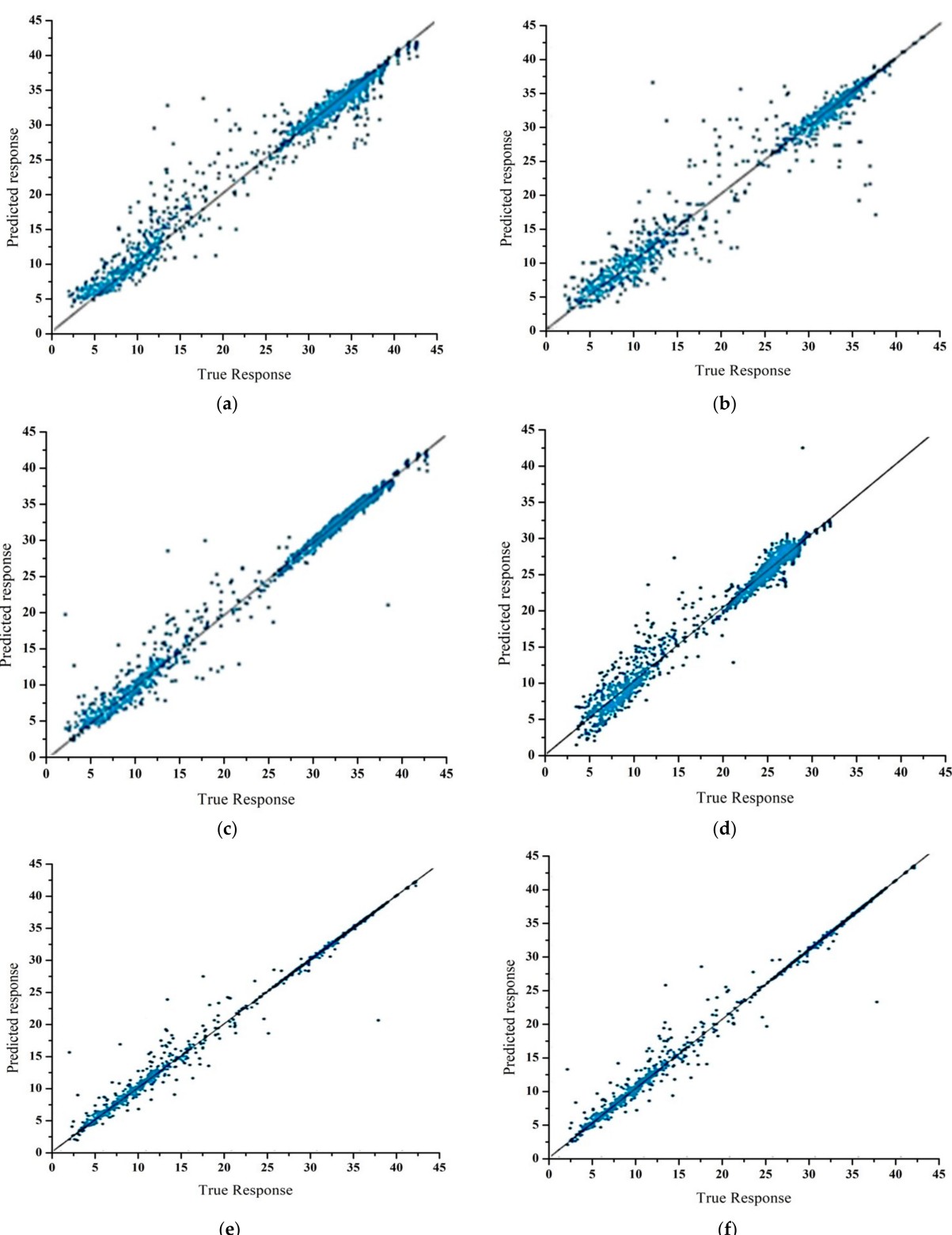

**Figure 8.** Regression models for mass flow rate (**a**) Ensembled Bagged Tree, (**b**) Tree Fine, (**c**) SVM Medium, (**d**) Stepwise Linear, (**e**) GPR Matern, and (**f**) GPR Rational Quadratic.

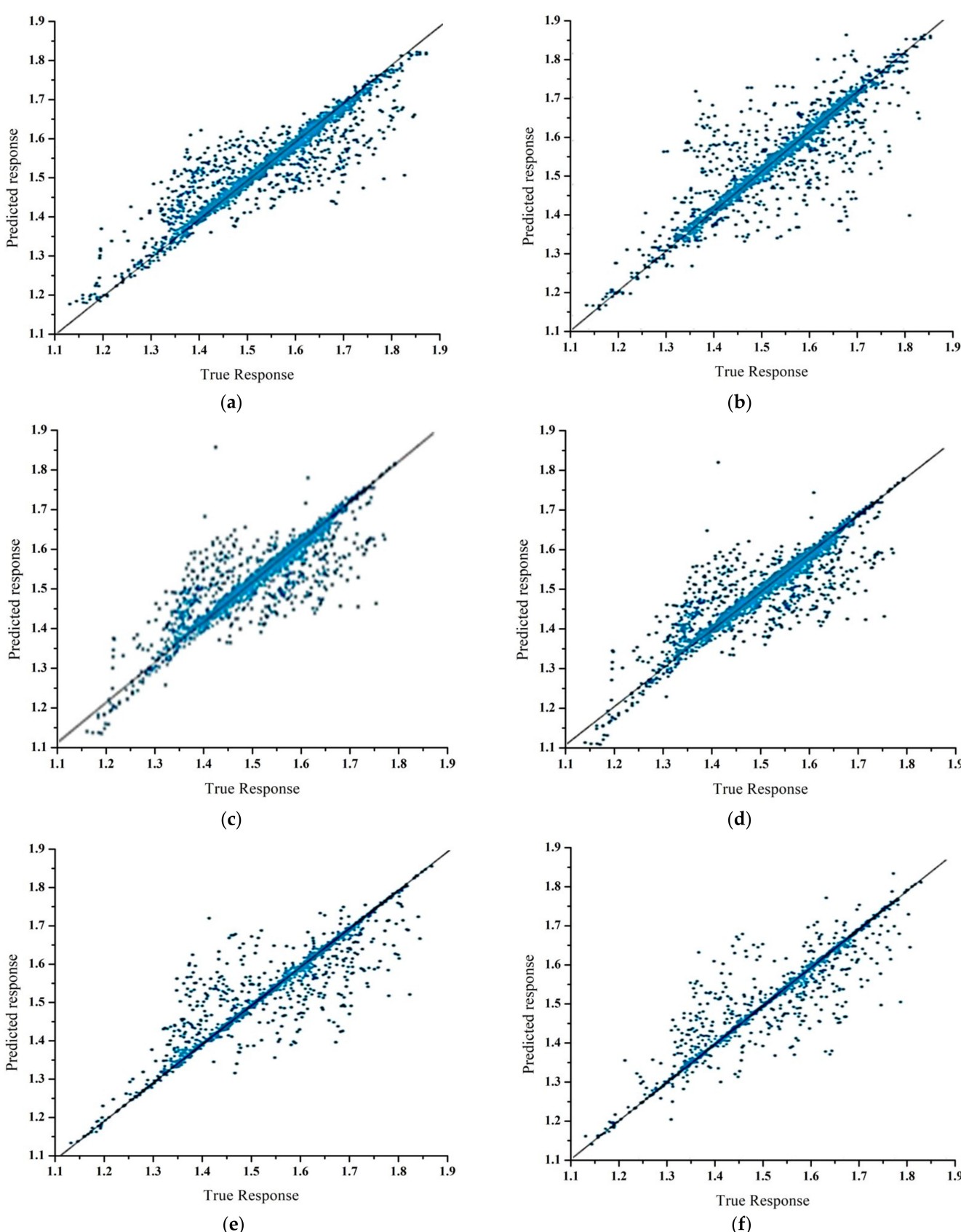

**Figure 9.** Regression models for pressure ratio (**a**) Ensembled Bagged Tree, (**b**) Tree Fine, (**c**) SVM Medium, (**d**) Stepwise Linear, (**e**) GPR Matern, and (**f**) GPR Rational Quadratic.

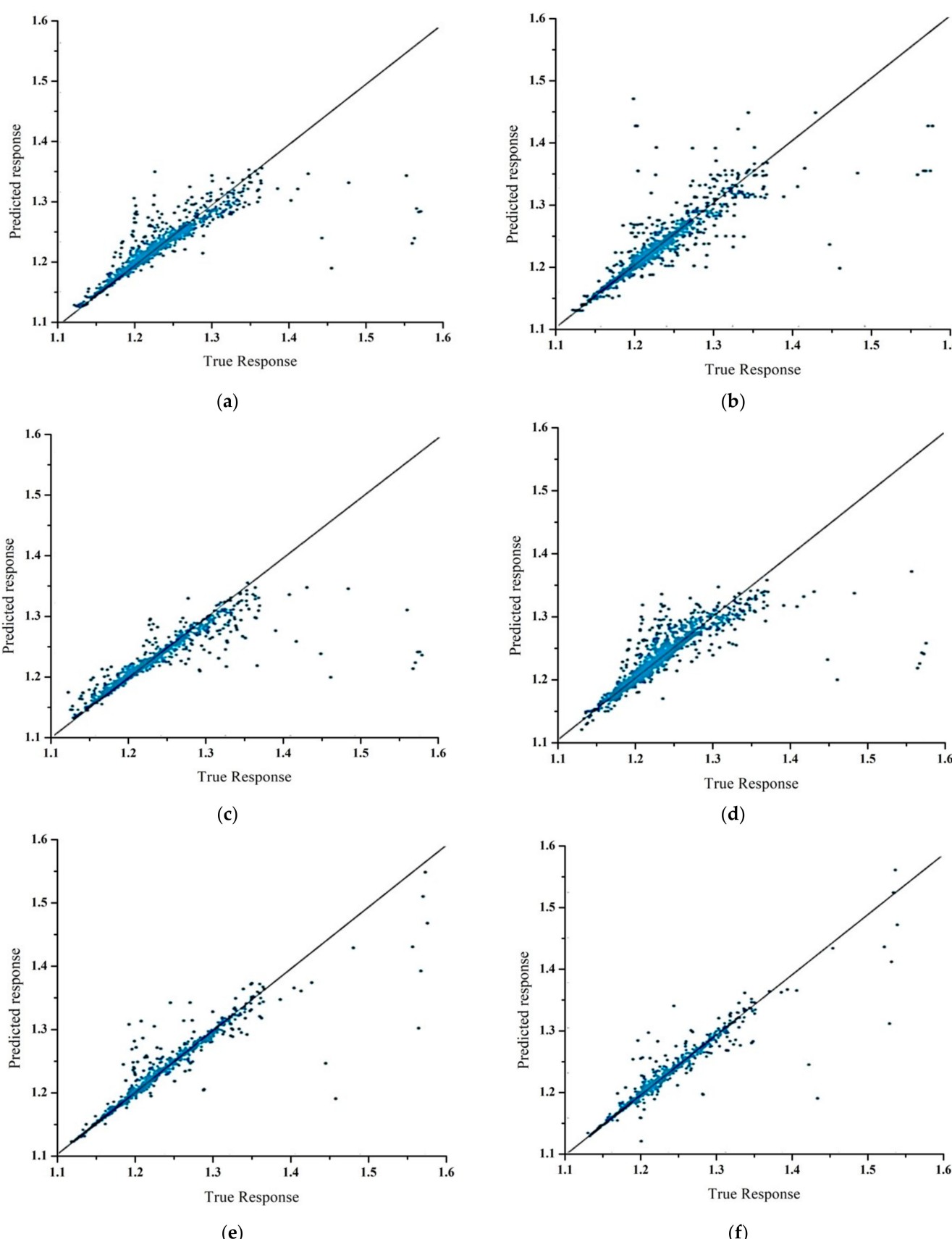

**Figure 10.** Regression models for temperature ratio (**a**) Ensembled Bagged Tree, (**b**) Tree Fine, (**c**) SVM Medium, (**d**) Stepwise Linear, (**e**) GPR Exponential, and (**f**) GPR Rational Quadratic.

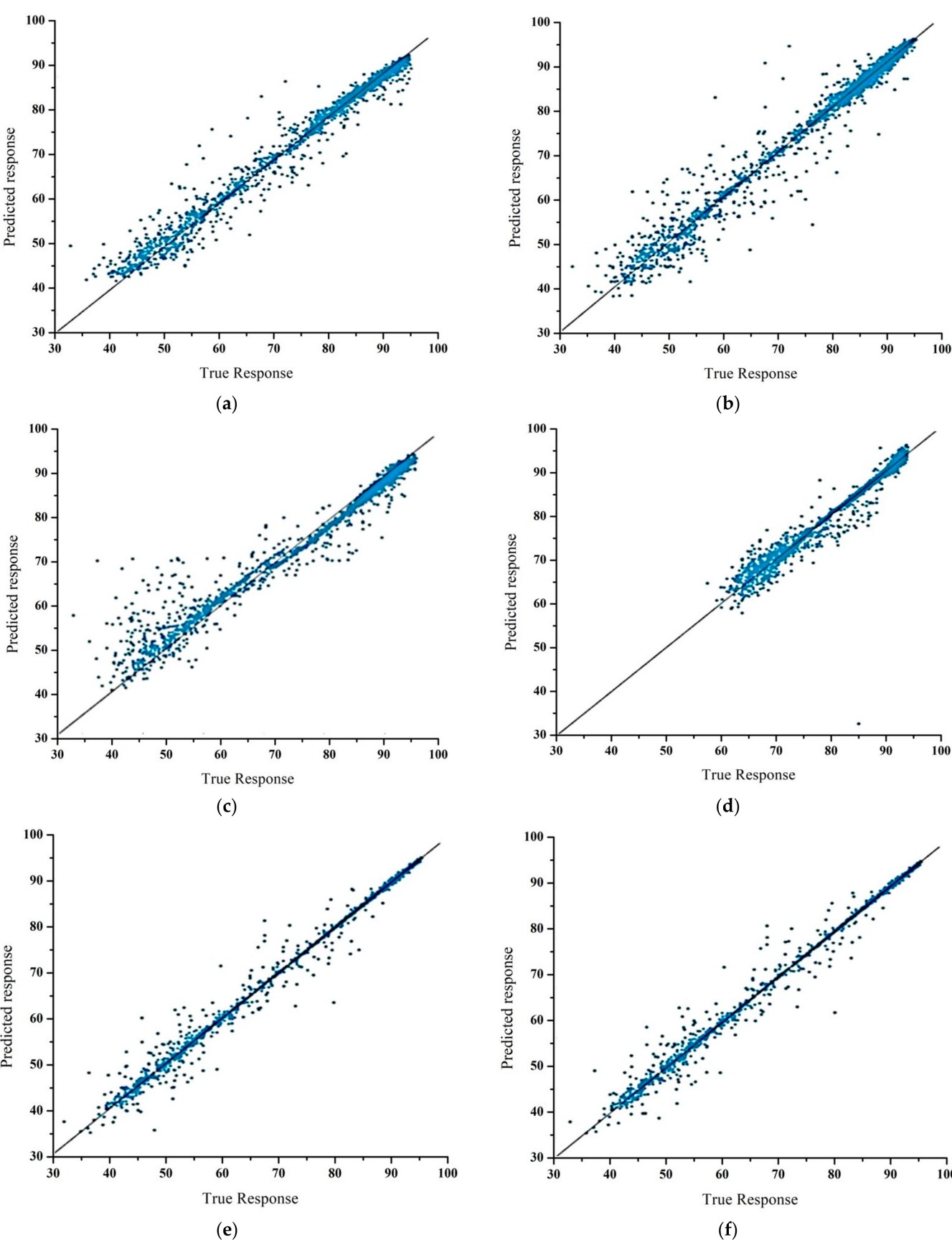

**Figure 11.** Regression models for isentropic efficiency (**a**) Ensembled Bagged Tree, (**b**) Tree Fine, (**c**) SVM fine, (**d**) Stepwise Linear, (**e**) GPR Matern, and (**f**) GPR Rational Quadratic.

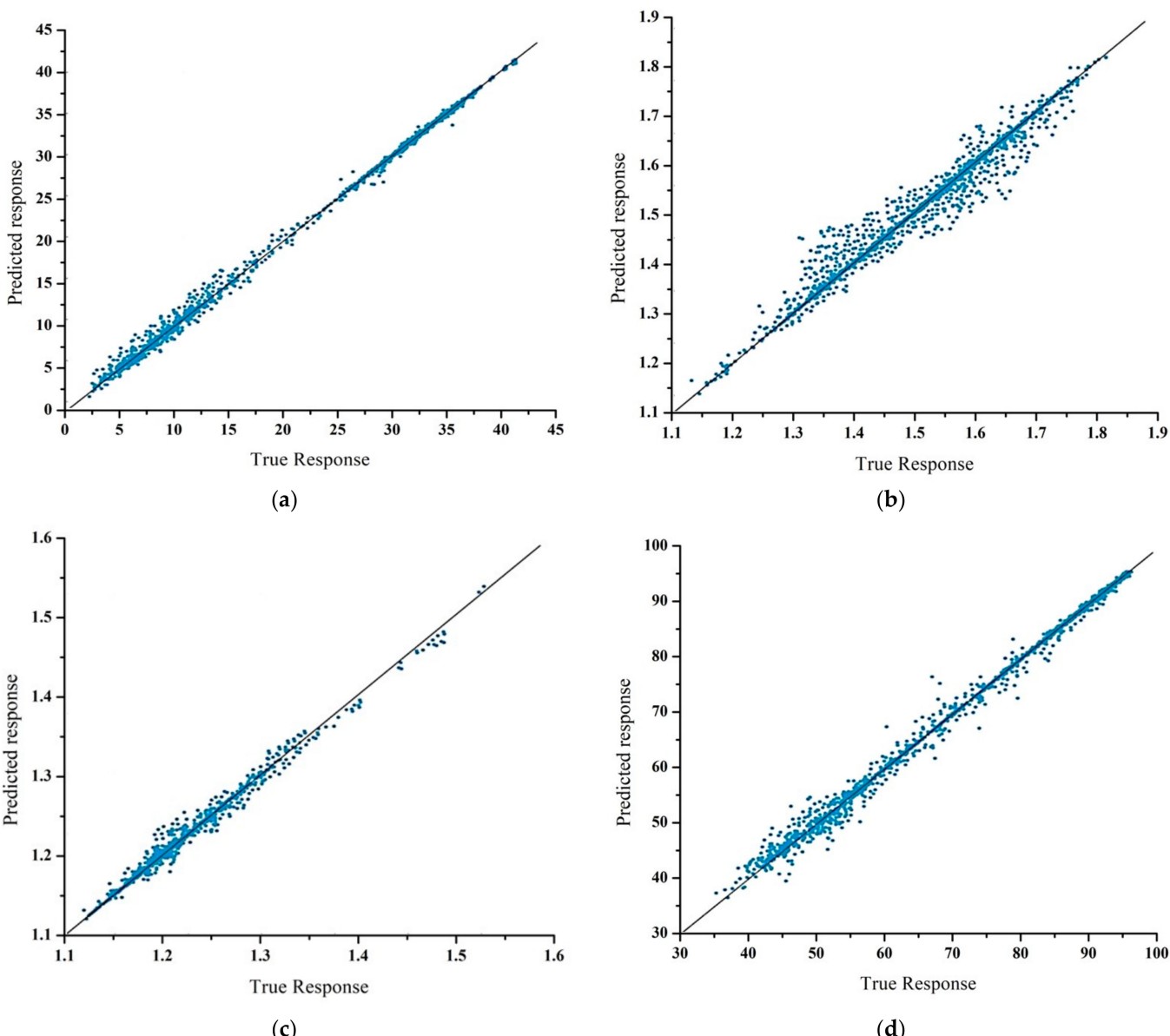

**Figure 12.** Best Response Regression Models (**a**) Exponential GPR for mass flow rate, (**b**) GPR Matern for pressure ratio, (**c**) GPR Matern for temperature ratio, and (**d**) GPR Exponential for isentropic efficiency.

On a 2.5-stage fully loaded axial compressor, a new Bezier surface modeling approach for the entire suction surface and pressure surface of blades was created, and the multi-island genetic algorithm was directly used for optimization. The smoothness of the blade surface can be ensured by the high-order continuity of control points [33]. The paper first adopted the novelty of extensive CFD analysis on distorted inlet flow, and then a different regression learner was applied to predict the compressor performance at variable conditions, as shown above. To achieve the optimum hyperparameters, two supervised learning methods, (SVR) and (GPR), had been used earlier to train the models with a Bayesian optimization algorithm. On five rotational speed lines, the qualified models are inserted into the through-flow code using the streamline curvature method (SLC) to estimate the overall output and internal flow field of the transonic compressor [34]. In the current research, sensitivity analysis has been applied to select the most influential features of the axial compressor at distorted inlet flow conditions and different RPM. The analysis had been conducted on linear regression, SVM, GPR, and tree regression learners. The Matern 5/2 kernel takes the spectral densities of the fixed component and applies Fourier

modifications to the Radial Basis function kernel. On the other hand, it does not measure data in high-dimensional spaces. GPR exponential and GPR Matern have greater statistical algorithms than other regression models, according to the results. The Gaussian Process Regression (GPR) models are difficult to understand, according to the findings.

Decision trees are non-parametric, interpretable, and fast, but they are locally optimized and prone to overfitting. Imbalanced classes also possess significant concern for decision trees. Although SVM is useful in high dimension space and acquires little memory, it is not suitable for large data sets, and target sets are closed to overlap. In stepwise regression models, predictive variables are carried out by removing the weakest correlated variable. Therefore, for case-sensitive datasets mostly, it is not suggested. The algorithm of Exponential GPR is indistinguishable from the Squared Exponential GPR. Aside from that, the Euclidean distance is not squared. Exponential GPR handles smooth capacities well with minor mistakes. However, it does not deal well with discontinuities. The Matern 5/2 kernel takes the fixed portion's spectral densities and makes Fourier changes to the Radial Basis function kernel. In contrast, it does not measure the data for high-dimensional spaces. Results show that GPR Exponential and GPR Matern have better predictive algorithms than other regression models. The Gaussian Process Regression (GPR) models have difficult interpretability. Still, they possess higher accuracy and are non-parametric Kernel-based probabilistic models, with multivariate distribution of the predetermined collection of random variables.

### 3.4. ML and DL Prediction Errors

Due to Single Independent Variables (SIV) levels and interactions with Multiple Independent Variables (MIV) in deep learning and machine learning, the effects of prediction analysis may differ slightly from an actual dataset. It is deduced that both machine learning and deep learning have produced predictions with exceptional accuracy based on the above statistical normalized RMSE and comparison between machine and deep learning with CFD results at different inlet conditions. Based on supervised learning, Table 6 shows the normalized RMSE calculations of ML and DL performance prediction for mass flow rate, pressure ratio, temperature ratio, and efficiency at Single Parameter Variable (SPV) and Multiple Parameter Variable (MPV) inlet conditions, respectively. In most cases, the results show an accurate prediction with an error of less than 1%, which is considered significantly reliable. The swirl flow in the compressor has intricate patterns, and its dataset is slightly above 1% error but in an acceptable range of deep learning prediction analysis. The result shows the comprehensive agreement of deep learning Artificial Neural Network multiple inlet parameters predictive results with CFD results analysis.

**Table 6.** Machine learning and deep learning RMSE comparison.

| Sr. No | Conditions | Norm. RMSE Mass Flow Rate | Norm. RMSE Pressure Ratio | Norm. RMSE Temperature Ratio | Norm. RMSE Efficiency |
|:---:|:---:|:---:|:---:|:---:|:---:|
| 1 | Analysis of deep learning $T_{amb}$ | 0.0033 | 0.0039 | 0.00071 | 0.00273 |
| 2 | Analysis of multiple independent variables using machine learning | 0.0089 | 0.0036 | 0.0041 | 0.0043 |
| 3 | Analysis of multiple independent variables using Deep Learning | 0.0259 | 0.0553 | 0.0091 | 0.018 |
| 4 | Analysis of multiple independent variables, excluding swirl flow using Deep Learning | 0.0218 | 0.0393 | 0.0081 | 0.0166 |

The Root Mean Square Error (RMSE) is frequently used to measure the difference between the predicted values of ML, deep learning analysis, and predicted CFD values. The individual differences between predicted and observed values are termed as residuals, whereas the RMSE aggregates these residuals into a single measure of predictive power. Figure 13a,b shows the RMSE comparison of the regression learner.

As discussed earlier, based on predicted versus actual results, which are dissipated close to the regression line, the response plot of these models is selected. Figure 13 shows that GPR Exponential has better RMSE results for isentropic efficiency and mass flow rate for the selection of the model. Similarly, for selecting pressure ratio and temperature ratio, GPR Matern has a better overall RMSE value, whereas GPR Exponential and GPR Quadratic lie close to the GPR Matern.

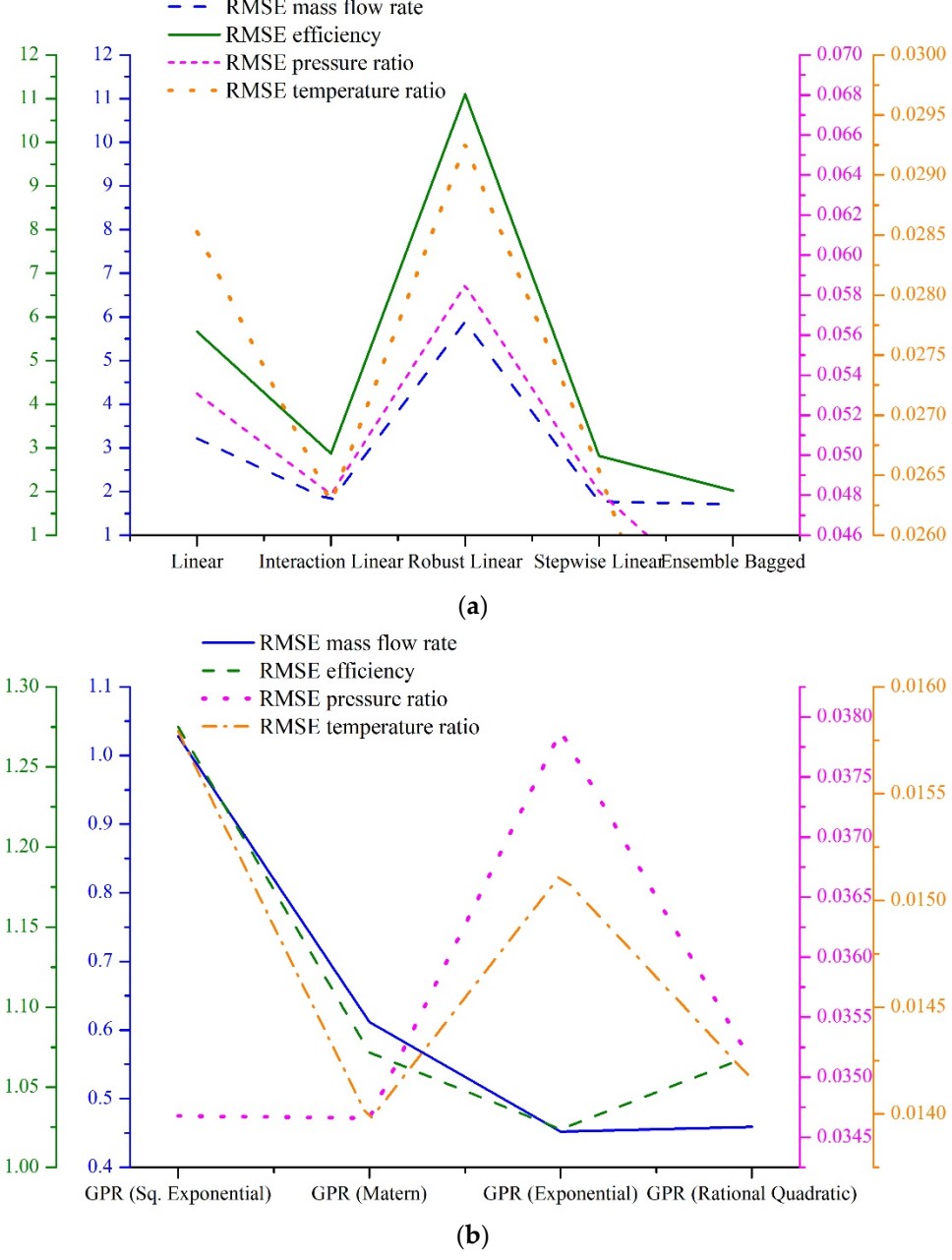

**Figure 13.** *Cont.*

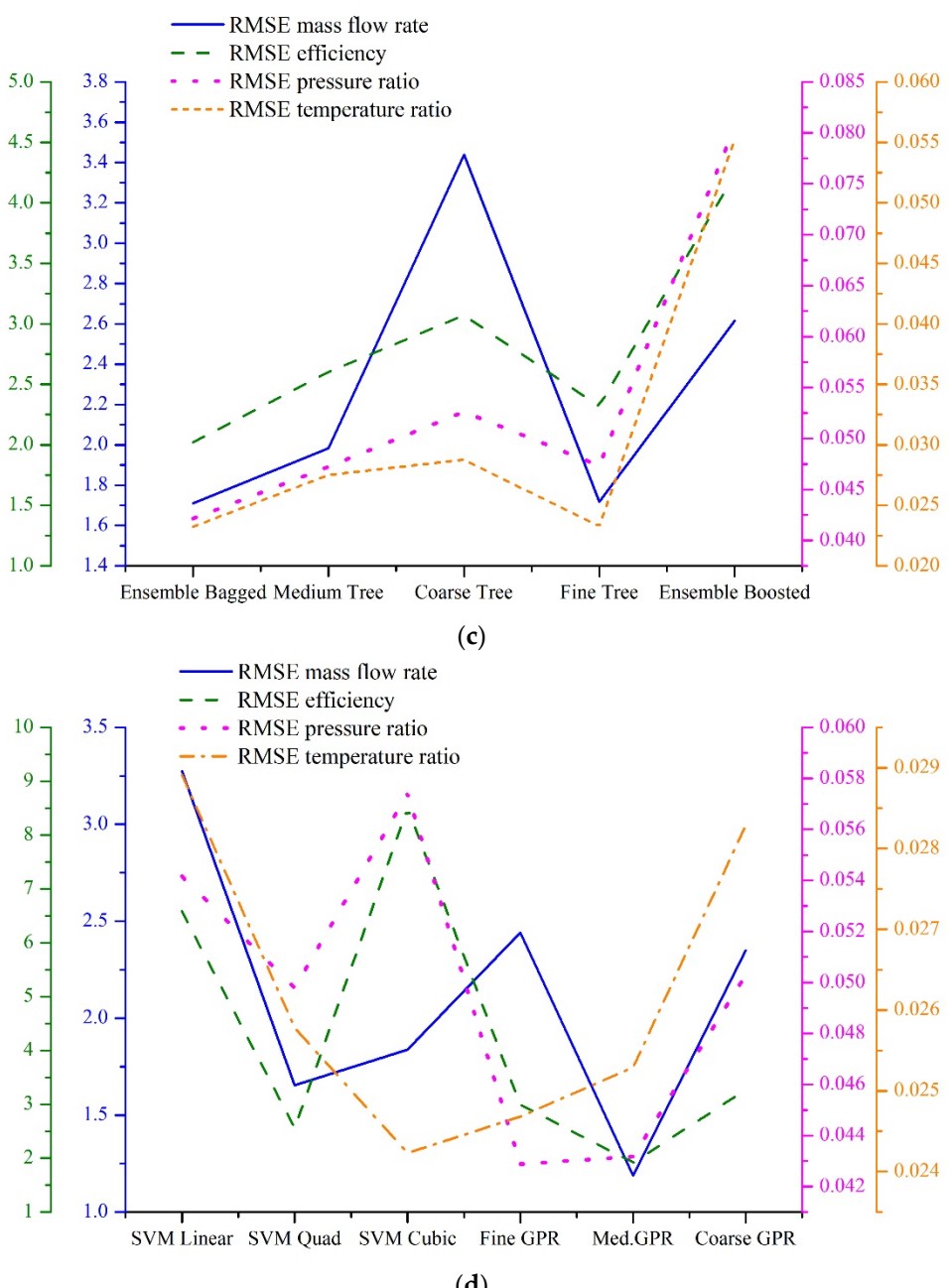

**Figure 13.** RMSE comparison (**a**) Linear RMSE, (**b**) GPR's RMSE, (**c**) Tree's RMSE, and (**d**) SVM's RMSE.

## 4. Discussion

The research emphasized numerical modeling, simulation, and prediction of transonic axial compressor rotor 67 performance at different inlet conditions. The paper focused on the utilization of regression models for different performance parameters through machine learning and Artificial Neural Networks. Initially, a computational fluid dynamics study was conducted under steady and distorted inlet flow conditions. Then, a supervised learning feed-forward network topology ANN model was applied to predict rotor-67 performance under different inlet steady and distorted conditions. In contrast with recent research, Sheng Qin et al. [35] proposed a multipurpose optimization approach based on reinforcement of the learning methodology. The hybrid optimization approach was applied for analysis of the total pressure drop and laminar flow field of a compressor cascade blade. Initially, the (Deep Deterministic Policy Gradient) DDPG network used an ANN-based

surrogate model as the environment. An approach based on reinforced learning multi-objective optimization was proposed, where the profile geometry was designated as the DDPG network's state, and the optimized geometry was considered as the action. Based on learned design experience, the optimizer approached fine performance cascade blade prediction. The introduction of novel objective functions over the pressure distribution of airfoils, such as the direction of the shock wave and a flat-roof-top factor, to design supercritical airfoils is a key component approach of A. Zeinalzadeh [36].

S. Pazireh and J. Defoe [37] introduced a blade profile loss model, which required the trailing edge boundary layer momentum thickness. An ANN was trained using over 400,000 variations of blade section form and flow conditions to approximate the momentum thickness for a given blade section. The training data were produced using a blade-to-blade flow field solver. Blade geometry and local flow conditions were the only parameters that the model considers. B.Cui et al. [38] coupled helicity-modified S–A model with a transition prediction model to improve the reliability and accuracy of the original S–A model for simulation of the transonic compressor rotor flows. The results show that modified helicity suppressed the strong vortex structures. Furthermore, the transition prediction model better analyzes the transition phenomena on both sides of the rotor blade.

Q. Cao et al. [39] investigated the performance deterioration of gas turbines and classified the major types of degradation i.e., by increasing tip clearance, corrosion/wear, fouling, and used Deep Neural Networks to forecast the degradation trend. By using a backpropagation algorithm optimized by Lenvenberg Marquardt, the researchers utilized the regression model to convert the efficiency and flow potential, which was determined by the thermodynamic model, into values under maximum load and ISO conditions. T. Olsson et al. [40] presented a novel data-driven approach based on real installation operational data for predicting the system degradation of micro gas turbines over time. A linear regression technique was employed for the estimation and forecasting of degradation.

## 5. Conclusions

CFD simulations are usually computationally expensive, require high memory, and are time-consuming iterative processes. In contrast, ML and deep learning have the most efficient approach to predicting the compressor's catastrophic failure and its stability for better performance at minimum computational cost and time. Flow physics, aerodynamics, and induced forces were analyzed. Different regression model approaches in ML and deep learning were used as prediction analysis for multiple independent variables simultaneously. The following conclusions can be drawn from the current work:

- Hub radial pressure distortion has improved the stability range of the compressor, whereas tip radial inlet flow distortion has deteriorated the compressor's performance.
- Detailed analysis of combined distortions in CFD analysis suggests that adding new distortion, i.e., either co-swirl flow or counter swirl flow in already existing inlet distortion, has a qualitative effect on the compressor. As distortion has a deteriorating effect on compressor performance, we cannot predict its impact on compressor stability.
- Different regression models in ML were used for prediction analysis of the compressor rotor performance. The results obtained from the different regression models depict a promising approach to predicting the compressor variables with minimum error. In contrast, the GPR algorithm was able to learn and trained the CFD-based dataset, thus providing promising results for prediction analysis.
- The results obtained from the Deep learning Artificial Neural Network (ANN) show that, despite using a conventional method of predicting parameters through CFD analysis, the use of ANNs is a promising approach to predicting the compressor's parameter rotor blade.
- The resulting regression learner and Artificial Neural Networks have a less than 1% prediction error at a computational cost, which is several times lesser than the underlying CFD solver's cost.

**Author Contributions:** Conceptualization, H.R.H. and K.P.; Formal analysis, M.U.S., H.R.H.; Investigation, M.U.S. and K.P.; Methodology, M.U.S., H.R.H.; Resources, A.M.K.; Software, M.U.S.; Supervision, H.R.H., K.P., A.M.K., Validation, M.U.S., K.P.; Visualization, H.E., M.K.; Writing original draft, M.U.S., U.A.; Writing review & editing, M.U.S., M.K., and A.I.; Result Discussion—M.U.S., A.I. All authors have read and agreed to the published version of the manuscript.

**Funding:** This research received no external funding.

**Institutional Review Board Statement:** Not Applicable.

**Informed Consent Statement:** Not Applicable.

**Data Availability Statement:** Not applicable.

**Acknowledgments:** I would like to thank Baoshan Zhu from Tsinghua University, China, and Andrew Ragai Anak Henry Rigit from UNIMAS, Malaysia expert reviews, valuable suggestions for the completion of the research. Furthermore, I also thank the Institute of Space Technology, Islamabad, for providing me computational resources and labs.

**Conflicts of Interest:** The authors declare no conflict of interest.

## Nomenclature

| | |
|---|---|
| Adam | Adaptive Moment Estimation |
| ANN | Artificial Neural Network |
| CFD | Computational fluid dynamics |
| CNN | Convolutional neural network |
| DL | Deep Learning |
| ISA | International Standard Atmosphere |
| KNN | k-Nearest Neighbors |
| ML | Machine Learning |
| MIV | Multiple Independent Variables |
| MSE | Mean Square Error |
| ReLU | Rectified Linear Unit |
| RPM | Revolution per minute |
| SVM | Support Vector Machine |
| GPR | Gaussian Process Regression |
| SIV | Single Independent Variables |
| TC | Tip clearance |
| $\mu$ | Sutherland viscosity |
| $\alpha$ | Flow Angle |
| Po | Total Pressure |

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
