# Peer review of "Prediction of Non-Uniform Distorted Flows, Effects on Transonic Compressor Using CFD, Regression Analysis and Artificial Neural Networks"

_applsci, doi:10.3390/app11083706_

Round 1

Reviewer 1 Report

The paper presents interesting study in topic of rapidly growing importance. It was interesting to read and could make a good research paper. There are three missing points which are critical in my opinion:

  • materials and methods section is chaotically organized
  • there are many technical issues that really affect the first impression (figures quality is very bad, use of other plots without displaying copyrights, little language and editorial issues)
  • paper does not include discussion of the results in the light of current state of the art

Materials and Methods

Section 2.1 structure is inconsistent. First the meshing procedure is described (ATM Optimized) in line 166. Then the reader needs to wait until line 192 to hear where the meshing procedure was performed. I would recommend restructuring the section to subsub sections or paragraphs devoting to: test stand, mesh (including mesh independence study), simulation definition, definition of distorted flow, dataset description. Structure could be different, but is indeed needed for clarity.

l. 155 - "Dataset is derived into 6-independent variables, i.e. (static pressure, RPM, ambient temperature, total pressure, and flow angularity at axial and radial flow) - where are those parameters measured that are required to predict compressor performance in terms of four dependent variables (mass flow rate, pressure ratio, temperature ratio, and isentropic efficiency)." - this sentence does not provide enough information about the datset. Where were the measurements taken? Was measurement pointwise or averaged on surfaces? Was it time-averaged?

l. 245 "To learn and better model the underlying distribution of input data, the samples have been split such that 80% of the simulation data comprises of training data. Table 3 shows the dataset division and the number of samples that fall in each category." - how were those 80% chosen?

Results

Figure 5 - description on meaning of the series names should be provided. One could try to guess them, but clarity is necessary.

Discussion

There is no discussion in this paper. One should expect the discussion to include features such as:

  • comparison of results with other studies
  • discussion of meaning of obtained results in the light of other studies
  • display novelty

Authors concentrated on explaining their own results without relating them to state-of-the-art including papers listed in the introduction as similar studies. There are numerous sources on the IMRAD paper structure that explain the importance and meaning of discussion of obtained findings (https://www.springer.com/gp/authors-editors/authorandreviewertutorials/writing-a-journal-manuscript/discussion-and-conclusions/10285528)

Language/Technical Issues

l. 182 "Total pressure has radial distribution in Hub and Tip." - why "Hub" and "Tip" are capitsalized. There are other places where inconsistent nomenclature is applied. Authors are sometimes capitalising names, sometimes not.

l. 208 "Designed Mass Flow Rate (kg/s)"

All Figures are of very bad quality. REsolution is very bad. Figures are not consistent  (f. eg. figure 7 versus figure 8). Figure 7 has non-white background. Axes annotation is inconsistent (different fonts, naming conventions - again figure 7 and 8 are a good example).

Some figures are copied from different papers without displaying copyright permission (fig. 1a, fig. 4)

l. 219 - "In this section, an artificial neural network model has been developed and trained for the same set of features required to perform the compressor simulation using CFD analysis, as shown in figure 2." - I guess this model was not "developed" in this section, but "described therein

"Evaluation Metric" is not bolded in line 249, "Model Architecture and Hyperparameters" as well. "Training of Neural Network" is bolded

l. 328 "Mach contours blade-blade validation" - do you meand "blade-to-blade"?

l. 458 GPR is explained after being used several times.

l. 476 "The individual difference between predicted and the observed values are termed as residuals"

Author Response

Dear Reviewer

            I am obliged and thankful for your timely response and action for the subject manuscript on behalf of all authors. I have revised my manuscript as per the reviewers' suggestions/comments. Kindly consider my following response to the reviewer's comments

All your suggestions have been addressed in attached file.

Regards

Reviewer 2 Report

Below you can find my comments for this re-submitted manuscript:

  1. The figures are still of bad quality, different font size are used, charts have different size, subsequent charts are arranged sloppily etc. (compare Fig 3a with Figure 3f, or Figure 9a with Figure 9b). I propose to re-do all the charts with a proper template to provide high-ended figures with the same size, line width, font, colour scheme etc.
  2. There is still no clarification if the computations were conducted for a single blade or a whole rotor. The text suggests the simulations were conducted for a whole rotor, however, the reference for your previous article [10] indicates the simulations were conducted for a single blade. This should be clarified.
  3. Line 191. “3D mesh at coarse, medium, fine, superfine and very-fine state is generated”, in table 2, authors mention Super Fine and Very Fine mesh. To be consistent I propose use consistent notation Super Fine and Very Fine
  4. Line 315, should be “Figure 3:”

Author Response

Dear Reviewer

            I am obliged and thankful for your timely response and action for the subject manuscript on behalf of all authors. I have revised my manuscript as per reviewers' suggestions/comments. Kindly consider my following response to the reviewer's comments

1          All figures have been updated and their quality has improved. The font size has been kept the same for all figures. All charts have been re-formed as suggested.

  1. Computations were conducted for a single blade, as the periodic condition was set for a whole rotor, as detail has been included in section 2.1.2
  2. Mesh table and text are now aligned accordingly.

Thanking You for your valuable suggestions

This manuscript is a resubmission of an earlier submission. The following is a list of the peer review reports and author responses from that submission.

Round 1

Reviewer 1 Report

Dear Authors,

the paper regards a complex approach to predict the influence of non-uniform distorted flow on the operation of a compressor rotor. The numerical model of a compressor rotor was validated against the NASA rotor 67 which is a well-examined benchmark for CFD studies. Mesh dependence studies were carried out. Further, the numerical studies were conducted for a wide range of operating conditions which differs by the one from each other by the pressure distortion, rotational speed etc. Based on a large data set obtained using ANSYS CFX, 2541 cases, the learning algorithms were employed to develop and train Artificial Neural Network. The Mean Square Error loss function was used to minimize the error of predicted data. Also, different regression models for Machine Learning were investigated. The paper is generally well written, with few typos, the main idea is clear and next steps are well explained, although there are some ambiguities in the CFD part. The figures are of bad quality, some of them are illegible, multiple fonts are used, I propose to re-do all the charts with a proper template to provide high-ended figures with the same size, line width, font, colour scheme etc.

I have some doubts regarding the numerical studies.

  • First of all the Authors should mention the employed wall treatment method and the y+ function value. It should be written clearly if the simulations were conducted for a single blade or a whole rotor. Right now the text suggests the simulations were conducted for a whole rotor, however, the reference for your previous article [11] indicates the simulations were conducted for a single blade. This issue is crucial and has to be clarified. I assume the simulations were conducted for a single blade, therefore I would like to point out that previous researchers indicate the importance of boundary layer computation and its crucial impact on the position of the shock wave and the boundary layer separation. Therefore, if the boundary layer is computed the influence of growth ratio for the following layers in the near-wall area should be examined. It is in major importance because you investigate the flow distortion in the radial direction, which will interact with the boundary layer and will affect the position of boundary layer separation and shock wave. Without a proper description of a boundary layer, the computation results are not reliable.

  • Secondly, the turbulent model choice is not fully understood for me, in the available literature, it is emphasised that for a simulation of a compressor rotor, excluding the LES, the SST model by Menter is the most reliable and recommended model ( M. Simoes, B. Montojos, N. Moura, J. Su, Validation of turbulence models for simulation of axial flow compressor, 20th International Congress of Mechanical Engineering November 15-20, 2009, Gramado, RS, Brazil; N. Spotts, X. Gao, A Comparison Study of Turbulence Models in RANS Simulations of Rotor 67, 54th AIAA Aerospace Sciences Meeting, January 4-8, 2016, San Diego, USA). Although I understand the investigation is focused more on the capabilities of Machine Learning algorithm, proving the usefulness and accuracy of machine learning on a data set that is not reliable is dubious.

Minor errors:

Line

  1. There should be no „.” at the end
  2. “3D mesh at coarse, medium, fine, and superfine state is generated.” In table 2, authors also mention Very Fine mesh. Moreover, to be consistent I propose to always start with the capital letter, i.e. “Fine”, “Super Fine” etc.
  3. “As flow solution is assumed to be periodic across the blade row, therefore, periodic boundary conditions were imposed on the top and bottom side of the blade domain.” I do not understand this sentence. Was the rotor divided into 22 parts and the simulations were conducted for a single blade? If so, the ~1 mln of elements mesh refers to a single blade? Do the top and bottom side refer to the hub and shroud or the blade-to-blade periodicity?
  4. Should be “Mesh dependence study”
  5. Authors sometimes write “fig. x”, sometimes “figure x”. I propose to use the same nomenclature in the whole paper.
  6. Figures captions are sometimes written as “Figure x.”, sometimes “Figure x:”. I propose to use the same nomenclature in the whole paper.
  7. The degree sign should be a superscript.
  8. The degree sign should be a superscript.

517: I would change the “Appendix A” with “Nomenclature”

Figure 2. The reference [15] in subfigure 2 (a) is wrong. You should refer to [12].

 Figure 3. I would advise preparing contours plots with constant step, i.e. 0.05 Mach. Having contours of numerical and experimental results with different step makes it difficult to compare. 

Figure 4. The subsequent figures of figure 4 should have the same size, lines width, font etc. I advise you to redo them with a proper chart template. I also advise using the same font on figures as in the text.

Figures 7-11. Axis titles and numbers are too small, therefore are not visible

Reviewer 2 Report

1) The principal drawback of the model is an employment of the obsolete k-e turbulence model (see line 167), which cannot produce up-to-date results of numerical modeling.

2) The flow in transonic compressor happens to be unsteady; that is why one should solve the time-dependent equations, not steady ones used in this paper, see line 166.

3) In line 167, the statement “RANS equations have been solved by using the k-e turbulence model” is stupid, as the equations can only be solved with a numerical method, not a turbulence model.

4) The dimension of the velocity components in line 176 is not explained.

5) A sketch of the computational domain indicating locations of the inlet and outlet must be given in Section 2.1 on page 4, not on page 8.

6) Figures have not been numbered in order of appearance, see lines 201, 208.

7) Figures 7-11 expose blind legends next to the coordinate axes.

8) There are many grammar errors (lines 2, 22-23, 35, 115, 126-128, 138,…), in particular, a multiple incorrect use of the word “Whereas”.